# Measurable Residual Disease Testing During Treatment with Bispecific Antibodies for Lymphoma

**DOI:** 10.3390/cancers17071153

**Published:** 2025-03-29

**Authors:** Gaston Jean-Louis, Hua-Jay J. Cherng

**Affiliations:** Division of Hematology/Oncology, Columbia University Irving Medical Center, New York, NY 10032, USA; gj2279@cumc.columbia.edu

**Keywords:** lymphoma, measurable residual disease, bispecific antibodies

## Abstract

Bispecific antibodies (BsAbs) have improved survival for patients with relapsed or refractory B-cell lymphomas, but some still experience disease progression. Predicting treatment response and durability remains a challenge. Measurable residual disease (MRD) has been studied as a biomarker to assess long-term outcomes and stratify patient risk after BsAb therapy. Recent studies show that MRD negativity correlates with better radiographic responses and longer progression-free survival. However, differences in MRD testing methods, timing, and technical aspects limit its current use in clinical practice. More research is needed to standardize MRD assessment before it can guide treatment decisions for B-cell lymphoma patients receiving BsAbs.

## 1. Introduction

Lymphoma is a heterogeneous group of malignant neoplasms that vary in clinical presentation and prognosis. Many aggressive lymphomas can be treated or cured with a combination of chemotherapy and immunotherapy. Treatment with the anti-CD20 monoclonal antibody rituximab has significantly improved outcomes for patients with both aggressive and indolent lymphomas [1]. The introduction of chimeric antigen receptor T-cell therapy (CAR-T), in which genetically engineered T-cells directed against antigens such as CD19 are used to generate immune-mediated anti-tumor activity, has also led to impressive treatment response rates in patients with relapsed and refractory (R/R) B-cell non-Hodgkin lymphoma (B-NHL) [2,3,4,5,6,7,8]. Long-term experience with the approved autologous anti-CD19 CAR-T cell therapies axicabtagene ciloleucel [9], tisagenlecleucel [10], and lisocabtagene maraleucel [11] confirms the curative potential of CAR-T in R/R aggressive large B-cell lymphomas (LBCL), including diffuse LBCL (DLBCL). Although early MRD analysis may help predict which patients may go on to experience durable remissions [12,13,14], unfortunately, less than half of patients with R/R LBCL will ultimately be cured by CAR-T; a cure is less likely in indolent lymphomas such as follicular lymphoma (FL) [15,16].

To add to the repertoire of T-cell-redirecting therapies, bispecific antibodies (BsAbs) that co-target endogenous immune effector cells and tumor-specific antigens have now been approved as an “off-the-shelf” immunotherapy treatment for R/R lymphomas. In addition to the need to address post-CAR-T relapses, logistical difficulties related to product manufacturing and access to accredited treatment centers [17], as well as concerns about treatment-related toxicity [18], indicate that BsAbs represent a vital alternative treatment modality.

## 2. Bispecific Antibodies

Blinatumomab, a BsAb against CD19 and CD3, was the first in this class to show significant efficacy in hematologic malignancies in a phase II trial evaluating its use in patients with relapsed/refractory B-precursor acute lymphoblastic leukemia [19]. The use of these drugs has expanded to B-NHL, with clinical trials evaluating the safety and efficacy of mosenutuzumab, glofitamab, and epcoritamab, as well as other similar products seeking regulatory approval. Approved BsAbs, or those in development for B-NHL, generally involve a fusion of two antibody moieties in an IgG-like structure that can simultaneously target CD20 on lymphoma cells and CD3 on endogenous T-cells; the co-localization of T-cells with B-NHL cells leads to immune effector cell activation and tumor cell killing through a major histocompatibility complex-independent mechanism [20].

Mosunetuzumab is a first-in-class CD20 × CD3 BsAb that was studied in patients with R/R B-NHL [21]. It demonstrated efficacy with an overall response rate (ORR) of 80% and a complete response (CR) rate of 60% in R/R FL after two or more prior therapies, with a median progression-free survival (PFS) of 24 months [22], leading to its Food and Drug Administration (FDA) approval for this indication. Epcoritamab, a bispecific antibody against CD3 and CD20, was evaluated in the phase I/II EPCORE NHL-1 trial [23] and is now approved for R/R LBCL [24] and R/R FL [25,26] based on efficacy in their respective disease cohorts. In LBCL, epcoritamab led to an ORR of 63% (CR 40%), a 24-month PFS of 28%, and a 24-month duration of CR rate of 64%. In FL, epcoritamab led to an ORR of 82% and a CR rate of 63%, with an 18-month duration of CR rate of 72%. Glofitamab, a BsAb with two binding sites for CD20 and one binding site for CD3, was evaluated in a phase I/II trial for patients with R/R LBCL [27,28], demonstrating an ORR of 52% and a CR rate of 39%, as well as a 12-month PFS of 37% and a 12-month duration of CR rate of 78%. Odronextamab is another CD20 × CD3 BsAb late in development [29], with particularly encouraging efficacy in FL [30]. The approvals of mosunetuzumab, epcoritamab, and glofitamab for R/R FL and/or LBCL have transformed the third-line treatment landscape for these malignancies; in particular, patients who experience CR have a high chance of durable remissions. However, predicting response and relapses remains a challenge, as the majority of patients with R/R B-NHL will still experience treatment failure, and the curative potential of BsAb is still unknown.

## 3. Better Response Assessment During T-Cell Redirecting Therapy

Assessing the response to lymphoma treatment, including after BsAbs, remains an ongoing area of research. Positron emission tomography/computed tomography (PET/CT) scans are generally used to evaluate the response to therapy and monitor for disease progression in lymphoma [31]. Despite encouraging CR rates for various indications, as described above, a significant fraction of complete responders will still experience future disease relapses, reflecting suboptimal PET/CT sensitivity for measurable residual disease (MRD) in this treatment setting. Additionally, specificity for disease activity versus treatment-related inflammation or pseudoprogression is an issue after both CAR-T [32] and BsAb [33,34]. The limitations of PET/CT amplify the need for alternative or complementary approaches to response evaluation and prediction of outcomes for patients who receive BsAbs and other T-cell redirecting therapies for lymphoma.

The assessment of MRD, which is the presence of malignant cells or tumor-associated molecules at a level below the threshold of detection of traditional disease monitoring methods, has increasingly been applied as a marker of treatment response for various hematologic malignancies, including lymphomas. MRD can be monitored to assess response to treatment, determine the risk of relapse, and predict the time to the next treatment after CAR-T therapy for lymphoma [12,13]. Several next-generation sequencing techniques have been used to assess MRD after the treatment of B-NHL with CAR-T cells and BsAb. The relevance of MRD in the context of BsAb is significant because of the potential long-term efficacy of this novel immunotherapy, particularly when treating curable aggressive lymphomas such as R/R LBCL. In this review, we will summarize MRD technologies in B-NHL, as well as the early data on MRD assessment and its prognostic utility during and after BsAb treatment.

## 4. MRD Testing Methodologies

Several strategies for MRD assessment have been developed to analyze specimens from various compartments. In some conditions, such as chronic lymphocytic leukemia (CLL), FL, and mantle cell lymphoma (MCL), circulating tumor cells (CTCs) can be directly assessed in the bone marrow or peripheral blood [35]. In these malignancies, multiparametric flow cytometry has been used to detect residual disease. Flow cytometry allows for the rapid detection of malignant cells, and the required instruments for this technique are generally available in clinical laboratories. This technique requires immediate sample processing and does not require a baseline tumor sample to measure a characteristic signal, but it generally has worse sensitivity compared to modern techniques [36].

Fragments of DNA exposed after the process of cell death can be detected in circulation as cell-free DNA (cfDNA). In the context of malignancies, cfDNA derived from tumors is referred to as circulating tumor DNA (ctDNA) [37]. For patients with lymphoma, ctDNA can be analyzed from samples obtained from accessible compartments such as plasma or cerebrospinal fluid to detect MRD. ctDNA is enriched within the pool of cfDNA in patients with aggressive lymphomas [38], making it a reliable biomarker in DLBCL. Because ctDNA has a relatively short half-life [39], it can be used for serial monitoring during treatments that can lead to rapid responses. Its relative abundance also makes it a richer source of tumor DNA than CTCs in DLBCL [40]. The primary methods of analyzing ctDNA or CTCs are polymerase chain reaction (PCR) and next-generation sequencing (NGS). Qualitative and quantitative PCR assays laid the foundation for MRD assessment in B-cell lymphomas. With sensitivities of ~0.005%, PCR-based assays are ideal for MRD monitoring in lymphomas with stereotyped genetic aberrations [35,41]. Allele-specific oligonucleotide (ASO)-PCR uses patient-specific primers for real-time quantitative PCR (RT-qPCR) analysis of targeted regions of interest. Digital droplet PCR (ddPCR) provides direct quantification of DNA after partitioning the sample to remove background noise from wild-type DNA. This method can also be multiplexed to analyze multiple targets [42]. While PCR-based techniques are relatively cost-effective, they assess fewer genomic targets and have lower sensitivity than NGS-based assays [35].

NGS can be used to detect multiple genomic alterations through massive parallel sequencing. NGS boasts higher sensitivity for MRD assessment compared to PCR-based assays due to its ability to detect many genetic aberrations simultaneously [43], thereby improving the utility of this strategy in patients with low tumor burden, such as those with early-stage disease or those who have received lymphoma-directed therapy. NGS-based methods for MRD detection generally have a longer turnaround time and higher cost compared to flow cytometry and remain less widely available [36].

The first NGS methodology to receive FDA approval for MRD monitoring in certain hematologic malignancies was clonotype sequencing, commercially available as clonoSEQ^®^. This method targets the rearranged V(D)J immunoglobulin receptor. A pretreatment sample is used to identify the dominant enriched clonotype, which is then measured in subsequent samples [40,44,45]. clonoSEQ can be applied to CTCs in peripheral blood or marrow tumor cells in CLL, MCL, ALL, or multiple myeloma. The application of this method to ctDNA has also been studied in DLBCL [39,40]. ClonoSEQ is currently commercially available for ctDNA sequencing in DLBCL but is not FDA approved for this indication.

Other methodologies for sequencing ctDNA from lymphomas include hybrid capture sequencing, in which a panel of probes is used to capture ctDNA libraries containing common tumor-specific mutations to allow for deep targeted sequencing [46,47]. Cancer Personalized Profiling by Deep Sequencing (CAPP-seq) uses a predefined panel of oligonucleotide probes to capture ctDNA libraries that overlap regions containing common mutations before sequencing. This approach improves sensitivity by focusing a high depth of sequencing on targeted regions [46,47]. Based on the CAPP-seq methodology, investigators have developed the AVENIO Oncology Assay Non-Hodgkin Lymphoma Test, which uses NGS of plasma cfDNA and matched genomic DNA to identify and monitor tumor-specific single-nucleotide variants in a panel of over 400 genomic regions [48]. A limitation of duplex sequencing techniques, such as CAPP-seq and related assays, is the requirement to detect tumor-associated single-nucleotide variants from both complementary strands of ctDNA, which can be challenging when there is limited input material. Phased variant enrichment and detection sequencing (PhasED-seq) is a next-generation assay that detects multiple mutations that commonly occur in close proximity in certain genomic regions in B-cell lymphomas. In this assay, these “phased variants” can be detected from single strands of ctDNA without duplex sequencing, using a hybrid capture panel for target enrichment [49], theoretically increasing the sensitivity for small amounts of MRD.

Whole genome sequencing (WGS), which offers greater “breadth” of sequencing and thus possibly greater sensitivity [50], has also been applied to ctDNA. Some studies have used low-pass WGS (lpWGS) in B-cell lymphomas to identify somatic copy number variants, which do not require deep sequencing for detection from plasma [51,52,53]. This technique thus offers a low-cost method of cfDNA analysis but may not be useful for MRD detection. Tumor-informed WGS involves the identification of patient-specific mutations from baseline tissue and the creation of a custom panel for MRD monitoring with deep sequencing, improving sensitivity but requiring a more labor-intensive workflow [54,55]. A summary of these methodologies is presented in Table 1.

## 5. MRD Assessments After BsAb Monotherapy for LBCL

MRD assessments have been included as secondary or exploratory endpoints in a number of trials investigating the use of BsAbs in pretreated B-cell NHL. Table 2 summarizes the results of the studies below. The most common use of MRD testing in studies of BsAbs in the literature to date is as a biomarker of response to therapy. In the phase I/II trial investigating epcoritamab in relapsed/refractory LBCL, Thieblemont and colleagues assessed MRD by ctDNA using clonoSEQ, starting at six weeks after treatment initiation [24]. With longer-term follow-up, a total of 54/119 (45.4%) of MRD-evaluable patients experienced MRD negativity after treatment, with 82.3% of these patients remaining MRD-negative at 6 months and 75.4% in a CR at 24 months. Most molecular responses occurred by cycle 3 day 1, and MRD negativity at this landmark timepoint was associated with improved PFS and overall survival (OS). Eleven patients with initial progressive disease on PET/CT after epcoritamab subsequently responded, with six experiencing durable CR and 5/6 demonstrating early MRD negativity, likely reflecting pseudoprogression [26]. The AVENIO ctDNA assay was also assessed and compared to clonoSEQ in this cohort, showing good concordance in ctDNA quantification but improved sensitivity. Baseline ctDNA levels by AVENIO were correlated with clinical characteristics, and MRD negativity (defined as <1 mutant molecule per cc) and major molecular response (≥2.5 log-fold reduction in ctDNA levels from baseline) by cycle 3 day 1 were associated with improved PFS [56].

In an exploratory analysis of glofitamab monotherapy in R/R LBCL, the reduction in ctDNA at cycle 3, as measured using the AVENIO assay, was greatest in patients who responded to treatment and was associated with PFS [57]. ctDNA status was particularly useful for stratifying outcomes for patients with a radiographic partial response at cycle 3 [58]. Baseline ctDNA levels measured using AVENIO were associated with classic prognostic factors and PFS, and patients with CR experienced sustained decreases in ctDNA levels until the end of treatment (EOT, 12 cycles). ctDNA sequencing could also be used to perform baseline liquid genotyping; patients with the LymphGen molecular subtypes BN2 or MCD [59] experienced the worst PFS [60]. Updated analysis with longer-term follow-up demonstrated that among patients who maintained a CR to EOT with glofitamab, 53 and 55% were MRD-negative by the AVENIO ctDNA assay at cycle 6 and EOT, respectively, and 72% had at least one MRD-negative assessment after EOT. An MRD-negative result, i.e., undetectable ctDNA, was defined by a *p*-value threshold of 0.005 [61].

ctDNA analysis in patients with R/R DLBCL treated with odronextamab monotherapy in the phase II ELM-2 study demonstrated that MRD negativity by a modified AVENIO ctDNA assay (*p* > 0.005) at cycle 5 day 1 was associated with significantly longer PFS. This included four patients who were not in a radiographic CR at that timepoint but subsequently converted. LymphGen classification based on baseline ctDNA analysis was possible, with the MCD subtype associated with worse PFS [62]. AZD0486 is a novel CD3 × CD19 targeting BsAb being investigated in an ongoing phase I study in R/R DLBCL. A total of 11/12 (92%) patients who experienced CR and were evaluable for MRD by the sensitive PhasED-seq assay had undetectable ctDNA [63].

Recent work has evaluated treatment with BsAbs in earlier lines of therapy for patients with LBCL who are ineligible for anthracycline-based chemotherapy regimens. The EPCORE DLBCL-3 phase II trial randomized 45 elderly patients with medical comorbidities and newly diagnosed LBCL to receive epcoritamab monotherapy, which was associated with an ORR of 74% and a CR rate of 64%. Of those who responded, 15 patients underwent AVENIO MRD testing, with 93% demonstrating MRD negativity [64]. 

**Table 2 cancers-17-01153-t002:** MRD assessments in trials of bispecific antibodies.

Trial	Trial Number	BsAb Regimen	Disease Indication	MRD Assessment Method	Definition of MRD Negativity	Key Findings	Reference
**BsAb Monotherapy for LBCL**							
**EPCORE NHL-1**	NCT03625037	Epcoritamab	R/R LBCL	clonoSEQ (ctDNA), AVENIO (ctDNA)	Undetectable ctDNA (clonoSEQ), <1 mutant molecule/cc (AVENIO)	45.4% achieved MRD negativity; MRD negativity at cycle 3 day 1 linked to PFS/OS; 82.3% remained MRD-negative at 6 months; pseudoprogression noted with early MRD negativity in 5/6 patients.	[24,26,56]
**Glofitamab Phase II**	NCT03075696	Glofitamab	R/R LBCL	AVENIO/CAPP-seq (ctDNA)	Undetectable ctDNA	MRD negativity correlated with PFS; sustained MRD negativity in CR patients until EOT; LymphGen molecular subtype MCD linked to poor PFS.	[57,58,59,60]
**ELM-2**	NCT03888105	Odronextamab	R/R LBCL	Modified AVENIO (ctDNA)	Undetectable ctDNA	MRD negativity at cycle 5 day 1 correlated with longer PFS; 4 patients not in CR at C5D1 later converted to CR.	[62]
**AZD0486 Phase I**	NCT05056727	AZD0486	R/R LBCL	PhasED-seq (ctDNA)	Undetectable ctDNA	92% of CR patients had undetectable ctDNA.	[63]
**EPCORE DLBCL-3**	NCT04628494	Epcoritamab	Newly diagnosed LBCL, elderly patients ineligible for anthracyclines	AVENIO (ctDNA)	Not specified	ORR 74%, CR 64%; 93% MRD-negative in responders.	[64]
**MRD in BsAb Combination Therapy Trials**							
**EPCORE NHL-2 Arm 1**	NCT04663347	Epcoritamab + R-CHOP	Newly diagnosed DLBCL, IPI 3-5, double-hit/triple-hit	AVENIO (ctDNA)	<1 mutant molecule/cc	91% MRD negativity; 83% achieved response by cycle 3 day 1.	[65]
**EPCORE-NHL2 Arm 8**	NCT04663347	Epcoritamab + R-mini-CHOP	Newly diagnosed DLBCL ineligible for R-CHOP due to age or comorbidities	AVENIO (ctDNA)	<1 mutant molecule/mL	MRD negativity of 95% in evaluable patients	[66]
**EPCORE NHL-5 Arm 3**	NCT04973137	Epcoritamab + Pola-R-CHP	Newly diagnosed LBCL	AVENIO (ctDNA)	<1 mutant molecule/cc	81% MRD-negative after 2 cycles; nearly all CR patients were MRD-negative at EOT.	[67]
**EPCORE NHL-5 Arm 1**	NCT04973137	Epcoritamab + Lenalidomide	R/R LBCL	AVENIO (ctDNA)	<1 mutant molecule/cc	74% of CR patients at cycle 2 were MRD-negative.	[68]
**COALITION**	NCT04851119	Glofitamab + R-CHOP or Glofitaamab + Pola-R-CHP	High-burden, high-risk LBCL	CAPP-seq (ctDNA)	Undetectable ctDNA (LOD 10^−4^)	88% MRD-negative at end of induction; MRD negativity increased after consolidation.	[69]
**NP40126**	NCT03467373	Glofitamab + R-CHOP	Untreated DLBCL	AVENIO (ctDNA)	Undetectable ctDNA	4/5 MRD-positive patients experienced a PFS event at 12 mos compared with 1/32 MRD-negative patients	[70]
**NP39488**	NCT03533283	Glofitamab + Pola	R/R LBCL	AVENIO (ctDNA)	Undetectable ctDNA	CR rate 62%, with mean duration of CR 31.8 months; all patients in CR had decreases in ctDNA	[71]
**BP41072**	NCT04077723	Englumafusp alfa + glofitamab	R/R B-NHL	AVENIO (ctDNA)	Undetectable ctDNA	Baseline ctDNA associated with pretreatment risk factors; used for baseline genotyping; molecular response correlated with radiographic response; better effector memory T-cell expansion associated with improved ctDNA response	[72,73]
**MRD in BsAb Therapy for FL and Other Lymphomas**							
**ELM-2**	NCT03888105	Odronextamab	R/R FL	AVENIO (ctDNA)	Undetectable ctDNA	MRD negativity at cycle 4 day 15 significantly correlated with PFS.	[30]
**EPCORE NHL-1**	NCT03625037	Epcoritamab	R/R FL	clonoSEQ (PBMC and ctDNA)	<1 cell per million nucleated cells or undetectable ctDNA	71% achieved MRD negativity; improved PFS in MRD-negative patients. ctDNA possibly more sensitive	[25]
**AZD0486 Phase I**	NCT04594642	AZD0486	R/R FL	PhasED-Seq (ctDNA)	Undetectable ctDNA	92% of CR patients had undetectable ctDNA within 12 weeks, 96% at any point	[74]
**NP30179**	NCT03075696	Glofitamab	R/R MCL	clonoSEQ (PBMCs or ctDNA)	Negative ctDNA	93% of evaluable patients had undetectable MRD	[75]
**EPCORE CLL-1**	NCT04623541	Epcoritamab	R/R CLL	clonoSEQ (PBMCs)	Undetectable at thresholds of 10^−4^ and 10^−6^	35% and 39% of patients achieved MRD negativity at 10^−4^ and 10^−6^, respectively	[76]
**Mosunetuzumab + Pola**	NCT05410418	Mosunetuzumab + pola	Untreated high tumor burden FL	PhasED-seq (ctDNA)	Undetectable ctDNA (10^−6^)	7/8 (88%) of patients with CR after cycle 2 had undetectable MRD	[77]
**EPCORE NHL-2 Arm 2**	NCT04663347	Epcoritamab + lenalidomide/rituximab	R/R FL + Lenalidomide/Rituximab	clonoSEQ (PBMC)	<1 cell per million nucleated cells	MRD negativity in 88% of evaluable patients associated with prolonged PFS.	[78]

*BsAb:* bispecific antibody. *MRD:* measurable residual disease. *R/R:* relapsed/refractory. *LBCL*: large B-cell lymphoma. *ctDNA*: circulating tumor DNA. *PFS:* progression-free survival. *OS*: overall survival. *CAPP-seq*: Cancer Personalized Profiling by Deep Sequencing. *LOD:* limit of detection. *CR*: complete response. *ORR:* overall response rate. *EOT:* end of treatment. *R-CHOP:* rituximab, cyclophosphamide, doxorubicin, vincristine, and prednisone. *Pola:* polatuzumab vedotin. *NHL:* Non-Hodgkin lymphoma. *FL*: follicular lymphoma. *MCL*: mantle cell lymphoma. *CLL*: chronic lymphocytic leukemia.

## 6. MRD Assessments After BsAb Combination Therapy for LBCL

MRD has also been evaluated as an endpoint in trials investigating the combination of BsAbs with other cytotoxic or novel agents. The EPCORE NHL-2 trial used the AVENIO assay to assess MRD as a secondary endpoint following treatment with epcoritamab and rituximab, cyclophosphamide, doxorubicin, vincristine, and prednisone (R-CHOP) as first-line therapy for LBCL with an International Prognostic Index (IPI) of 3–5 or double-hit status (containing rearrangements of MYC and either BCL2 or BCL6). This combination was associated with MRD negativity in 30/33 (91%) of MRD-evaluable patients, with 83% of MRD-negative patients experiencing a molecular response by cycle 3, day 1 [65]. A separate arm of EPCORE NHL-2 evaluated epcoritamab with R-miniCHOP for frailer or older patients and noted an MRD negativity rate of 20/21 (95%) at any point in evaluable patients [66]. The EPCORE NHL-5 trial explored the addition of epcoritamab to polatuzumab vedotin (pola) and R-CHP and also evaluated ctDNA kinetics by the AVENIO assay; it demonstrated that 81% of evaluable patients were MRD-negative after 2 cycles, and all but one patient with CR experienced MRD negativity by EOT [67]. Another arm of EPCORE NHL-5 investigated the combination of epcoritamab with lenalidomide in R/R LBCL and noted sustained drops in ctDNA levels throughout treatment, with 14/19 (74%) of evaluable patients in CR after 2 cycles experiencing MRD negativity [68]. These studies all defined “MRD-negative” as < 1 mutant molecule per cc, though the lower limit of detection of the AVENIO assay for ctDNA is lower than this threshold.

The COALITION study is a randomized phase II trial comparing glofitamab with R-CHOP versus pola-R-CHP in patients younger than 65 with high-risk newly diagnosed LBCL. Patients received an additional two cycles of consolidation glofitamab after the conclusion of combination therapy and were profiled using a CAPP-seq based ctDNA assay with a described lower limit of detection of 10^−4^. A total of 50/57 (88%)patients were MRD-negative at the end of induction, including 8/10 patients with a partial response at this time point. ctDNA levels continued to decrease or convert to MRD negativity after glofitamab consolidation [69]. A separate study of glofitamab added to R-CHOP (NP40126) demonstrated that 4/5 patients who were MRD-positive by AVENIO ctDNA analysis at EOT experienced a PFS event by 12 months, while 1/32 who were MRD-negative experienced a PFS event in the same timeframe [70].

Glofitamab has also been studied in combination with polatuzumab vedotin. In an extended follow-up of a phase Ib/II trial of this combination for patients with heavily pretreated R/R LBCL, a CR rate of 62% was observed, with a median duration of CR of 31.8 months. Notably, a reduction of ctDNA measured by AVENIO was observed in all patients with CR at the end of treatment, regardless of LBCL histology [71]. Englumafusp alfa is a novel CD19 and 4-1BBL fusion protein that acts as a co-stimulatory molecule in the context of T-cell activation. When combined with glofitamab for R/R B-NHL in a phase I dose-escalation study, it led to an ORR of 67% and a CR rate of 57% in aggressive histologies. Baseline ctDNA, as measured using the AVENIO assay, was associated with pretreatment risk factors and could be used to perform baseline genotyping, and molecular response by cycle 3 day 1 was significantly associated with radiographic response; the MRD negativity rate was 32% at that time point. Better effector memory T-cell expansion was associated with improved ctDNA response by EOT [72,73].

Whether MRD testing can be used to guide changes in treatment is another area of active investigation. A reduction in ctDNA levels during first-line therapy for LBCL is associated with improved survival [79]. Falchi and colleagues investigated whether early intervention for patients with elevated ctDNA levels during first-line therapy improves outcomes for patients with LBCL. Patients with LBCL underwent AVENIO ctDNA assessment after one cycle of first-line R-CHOP and were defined as high risk if they did not experience a 2 log-fold reduction in ctDNA. A total of 29 high-risk patients continued R-CHOP through cycle 6 and received glofitamab starting in cycle 3. The investigators found that the combination of glofitamab with R-CHOP in these high-risk patients was associated with a favorable EOT ORR and CR rate of 95% and 85%, respectively [80]. These data suggest that interim MRD testing may be useful in guiding treatment intensification.

## 7. MRD Testing After BsAb Treatment for FL or Other Lymphomas

The use of MRD assessments as predictors of clinical outcomes is similarly being evaluated in trials of BsAbs for other R/R B-NHL. The ELM-2 trial evaluated the CD20 × CD3 BsAb odronextamab in patients with R/R FL (N = 128). MRD assessment was performed using NGS ctDNA analysis with the AVENIO platform at baseline and on cycle 4, day 15. MRD clearance was defined based on the assay’s limit of detection (*p* > 0.005). There were 64 MRD-evaluable patients with paired baseline and mid-treatment samples; PFS was significantly improved if the patient was MRD-negative at the landmark timepoint (HR 0.26). Of the 14 patients who discontinued odronextamab for disease progression, none experienced MRD negativity at any point [30]. There was a trend toward improved PFS for patients with a CR by PET/CT at this same timepoint if they were MRD-negative versus MRD-positive (*p* = 0.072). TP53 mutation was identified from ctDNA in 34/53 (64%) evaluable patients at baseline and was associated with a higher chance of interim MRD positivity and shorter PFS in a prior analysis [62].

Patients with R/R FL treated with epcoritamab in the pivotal dose phase II expansion cohort of EPCORE NHL-1 were analyzed for MRD using the clonoSEQ assay performed on peripheral blood mononuclear cell (PBMC) samples. Samples were collected on day 1 of cycles 3, 5, 7, 10, and 13, and every 6 months during follow-up. A total of 91/128 (71%) were MRD-evaluable, and 61/91 (67%) experienced MRD negativity at any point. PFS was improved across the cohort, including in patients with bulky disease, those with ≥3 prior lines of therapy, and those with disease progression within 24 months of initial immunochemotherapy who experienced MRD negativity, as well as from cycle 3 day 1 in a landmark analysis [25]. When including the cycle 1 optimization cohort, which used a different ramp-up dosing schedule, the pooled MRD negativity rate for 135 evaluable patients was 89/135 (66%) [81]. An exploratory analysis compared MRD analysis in R/R FL from EPCORE NHL-1 using clonoSEQ sequencing on PBMCs and ctDNA. The MRD negativity rate from ctDNA in 100 evaluable patients was 68%. A total of 338/414 (82%) matched samples were concordant in MRD status between PMBCs and ctDNA, and for 89 unique patients with both assays performed, 79/89 (89%) had concordant MRD status. MRD negativity was defined as < 1 cell per million nucleated cells (corresponding to a sensitivity of 10^−6^) for PMBCs and 0 clones per 1 cc of plasma for ctDNA. It took a median of 28 and 57 days to experience MRD negativity when analyzing PBMCs and ctDNA, respectively, possibly reflecting differences in sensitivity. Patients with discordant MRD status between PBMCs and ctDNA experienced poorer PFS compared to MRD “double negative” patients [82]. AZD0486, a novel IgG4 fully humanized CD19 × CD3 BsAb, was also tested in R/R FL in a phase I study. In this trial, CR rates ranged from 75 to 88% depending on the target dose. Of the 26 patients who received a target dose of ≥2.4 mg of AZD0486 and experienced a CR, 24/26 (92%) had undetectable ctDNA within 12 weeks, and 25/26 (96%) at any point [74].

BsAbs have also been tested in other B-NHL histologies, with correlative analysis of MRD. In patients with R/R MCL treated with glofitamab in the NP30179 phase I/II study, 14/15 (93%) of evaluable patients had undetectable MRD by clonoSEQ on PBMCs or ctDNA by C3 [75]. Patients with R/R CLL treated with epcoritamab in the EPCORE CLL-1 dose expansion and cycle 1 optimization cohorts were profiled for MRD by clonoSEQ on PBMCs. Of the 23 patients evaluable for MRD at any point, 8 and 9 (35 and 39%) experienced undetectable MRD at the 10^−4^ and 10^−6^ thresholds, respectively, with higher molecular response rates in patients with CR [76].

Other trials have evaluated the use of MRD testing in the context of novel combination therapy with BsAb for FL. In a phase II investigator-initiated trial of mosunetuzumab and polutuzumab vedotin for previously untreated FL, MRD was assessed from ctDNA using PhasED-seq in the first 11 patients enrolled. Of the eight patients with CR by PET/CT after cycle 2, seven had achieved undetectable MRD at the level of 10^−6^ [77]. The combination of fixed-duration epcoritamab with lenalidomide and rituximab in the R/R FL EPCORE NHL-2 study led to an impressive 21-month PFS rate of 86% in 66 of 75 (88%) MRD-evaluable patients who were MRD-negative by clonoSEQ on PBMCs at any point; this compared favorably to a 21-month PFS rate of 44% in the minority of patients who did not experience MRD negativity [78].

## 8. Barriers and Future Directions

Although the aforementioned studies represent important advancements in the incorporation of molecular response and MRD assessment in evaluating the efficacy of BsAb in the treatment of R/R B-NHL, challenges remain in interpreting these findings. Standardization must be a priority; several different assays (clonoSEQ, CAPP-seq-based, and PhasED-seq) with variations in technical performance/limit of detection, units of measurement, and even the peripheral compartment analyzed have been utilized. “MRD negative” has not consistently equated to undetected MRD in all studies, and even across studies using the same assay, there has been heterogeneity in how molecular response thresholds are defined. Technical information on assays can be missing in abstract presentations or manuscripts with limited space for the description of correlative studies. Sampling methodology can be quite important for sensitivity for MRD, as storage temperature [83] and collection tubes can affect sample quality. While cfDNA-stabilizing tubes and tubes containing the preservative ethylenediaminetetraacetic acid (EDTA) perform similarly at short interval time points, detectable ctDNA can decline by approximately 50% in EDTA tubes after 48 h if not processed [84]. “MRD evaluable” patients invariably represent a subset of the total treated cohort, likely because of missing samples or other technical or logistical limitations; however, when molecular response is only described in patients with the best radiographic response to treatment, this further complicates the interpretation of study results. The timing of interim MRD measurement (after cycle 2, after cycle 4, or MRD-negative at any point) has varied across trials as well. As was the case for PET/CT imaging in lymphoma a decade ago [31], further work is needed to unify approaches to MRD assessment in the context of BsAb treatment.

The National Comprehensive Cancer Network (NCCN) guidelines for B-cell lymphomas [85] recently added a category 2B recommendation to perform ctDNA MRD assessment to help adjudicate equivocal EOT PET/CT results after frontline immunochemotherapy, using an assay with a limit of detection of < 1 part per million. Although this does not pertain specifically to BsAb treatment in R/R B-NHL, it represents an important introduction of formal guidelines for MRD assessment. As the treatment options for B-NHL continue to expand, collaboration between investigators and industry is needed to advance MRD assessment beyond being simply an exploratory endpoint in individual studies. Cross-trial data sharing and real-world studies of MRD assessment in patients treated with standard-of-care BsAb treatment could increase cohort size and statistical power for determining the prognostic utility of MRD assessment, while workshops and working groups [86] could help coalesce our approach to MRD assessment timing, preferred assays, and definitions of MRD negativity.

In addition to serving as a prognostic biomarker, in the future, MRD could be used to guide BsAb treatment decision-making, though additional prospective studies are needed. In addition to the intensification of R-CHOP with glofitamab in patients with suboptimal early molecular response on the AVENIO ctDNA assay [80], additional MRD-adapted BsAb studies in LBCL are ongoing or planned. These include EpLCART (NCT06414148), which investigates the addition of epcoritamab monotherapy or its combination with lenalidomide and rituximab in MRD-positive patients after standard anti-CD19 CAR-T, GRAIL (NCT06050694), where patients with unfavorable ctDNA response after cycle 1 of pola-R-CHP and/or PET/CT response after cycle 2 will receive the addition of glofitamab, and NCT06670105, where patients who are MRD-positive by the PhasED-seq assay after standard frontline immunochemotherapy will receive glofitamab. In the planned GOLD trial, older patients with DLBCL who achieve CR by PET/CT but have detectable ctDNA by clonoSEQ after pola-R-mini-CHP will receive consolidation therapy with mosunetuzumab, while patients with undetectable ctDNA will undergo observation (NCT06828991). NCT06682130 uses MRD assessment to guide the use of autologous stem cell transplant (ASCT). Patients with R/R DLBCL who have a partial response or ctDNA positivity after second-line salvage will undergo ASCT or receive glofitamab as bridging therapy to ASCT. Patients with CR and ctDNA negativity after second-line treatment will receive ASCT as consolidation immediately. These trials are summarized in Table 3**.** One can also imagine studies in which patients with suboptimal interim molecular response to BsAb monotherapy could escalate to combination therapy with novel agents. Liquid genotyping for patient selection based on genomic/molecular features in ctDNA could also be incorporated into study design. MRD-adapted approaches have the potential to personalize BsAb treatment according to individual patient response.

The possibility of tailoring treatment duration could be especially valuable when comparing fixed-duration and continuous therapies. It remains an open question whether epcoritamab could be safely discontinued in MRD-negative patients and re-initiated upon MRD conversion, a strategy that would require prospective validation. Conversely, it is unknown whether patients who have detectable MRD at the end of treatment with glofitamab would benefit from the continuation of therapy. The approach of using MRD testing to shorten maintenance therapy was evaluated for rituximab maintenance in advanced FL in the FOLL12 Study, although PFS in patients receiving an MRD-adapted approach was inferior to that of those receiving standard rituximab maintenance [87]. Current recommendations for mosunsetuzumab indicate that patients with a partial response or stable disease after eight cycles of therapy should receive an additional nine cycles. However, further investigation should be conducted to determine whether mosunetuzumab can be discontinued in patients who achieve MRD negativity but non-CR on imaging. Using MRD negativity to guide therapy duration could prolong survival, lower overall treatment costs, reduce the risk of infections and other adverse events, and improve quality of life by decreasing the treatment burden.

## 9. Conclusions

MRD assessment is emerging as a critical tool in the management of lymphoid malignancies, offering prognostic information that could one day help guide treatment planning. Here, we have summarized the next-generation sequencing methods of MRD assessment utilized in clinical trials evaluating BsAb single-agent and combination treatment for newly diagnosed and R/R B-NHL. When evaluating MRD as a biomarker of response, multiple studies have demonstrated that achieving MRD negativity correlates with superior PFS, or even OS. A growing collection of data suggests that MRD kinetics during early treatment cycles may serve as an early indicator of long-term outcomes. In some cases, liquid genotyping was used to identify high-risk molecular subtypes with suboptimal treatment response. Whether on-treatment MRD status can be used to adapt treatment remains to be seen, and further investigation of this possibility will be critical. Challenges still remain regarding the standardization of approaches to MRD testing, as well as consolidating data and findings across studies. Further work is required to address these barriers to consensus practices. As BsAbs and other T-cell redirecting therapies continue to reshape the treatment landscape for lymphoma, MRD testing may soon play a critical role in response assessment, risk stratification, and personalized treatment planning.

## Figures and Tables

**Table 1 cancers-17-01153-t001:** MRD detection assays.

Assay Type	Assay Name(s)	Methodology	Unique Advantages
Allele-specific oligonucleotide PCR		qRT-PCR with patient-specific primers	Ease of use, high sensitivity
ddPCR		Direct quantification of DNA after sample partitioning	Quantitative, high sensitivity, allows for multiplexing
Flow cytometry		Multiparametric cell analysis to detect abnormal immunophenotypes of cells in peripheral blood	Fast turnaround time, widely available in clinical laboratories, no baseline tumor sample required
Clonotype sequencing	clonoSEQ	Identification and tracking of unique VDJ rearrangement	Commercially available, for PMBCs and cfDNA
Targeted hybrid capture	CAPP-seq, AVENIO NHL	Hybrid capture of lymphoma-specific panel for deep duplex sequencing	High sensitivity, liquid genotyping possible
Phased variant hybrid capture	PhasED-seq	Hybrid capture of areas enriched mutations in close proximity (phased variants) for deep non-duplex sequencing	Ultra-high sensitivity
Low-pass whole genome sequencing		Low coverage sequencing to detect copy number variants	Simple workflow, low cost
Tumor-informed whole genome sequencing		Identification of patient-specific mutations from baseline malignant tissue	Personalized MRD monitoring, improved sensitivity

qRT-PCR: quantitative real-time polymerase chain reaction. ddPCR: digital droplet PCR. PMBCs: peripheral blood mononuclear cells. cfDNA: cell-free deoxyribonucleic acid. CAPP-seq: Cancer Personalized Profiling by Deep Sequencing. AVENIO NHL: AVENIO Oncology Assay Non-Hodgkin Lymphoma Test. PhasED-seq: Phased variant enrichment and detection sequencing.

**Table 3 cancers-17-01153-t003:** Open or planned trials of MRD-tailored bispecific antibody therapies for lymphoma.

Trial	Trial Number	BsAb Agent	Indication	Use of MRD
	NCT04980222	Glofitamab	1L LBCL	BsAb intensification for patients with unfavorable ctDNA response after C1 of R-CHOP
EpLCART	NCT06414148	Epcoritamab +/− lenalidomide and rituximab	R/R LBCL	BsAb treatment if MRD-positive after anti-CD19 CAR-T
GRAIL	NCT06050694	Glofitamab	1L LBCL	BsAb intensification for patients with unfavorable ctDNA response after C1 of Pola-R-CHP
	NCT06670105	Glofitamab	1L LBCL	BsAb consolidation if MRD-positive after standard 1L treatment
GOLD	NCT06828991	Mosunetuzumab	1L DLBCL	BsAb consolidation if MRD-positive after Pola-R-mini-CHP
	NCT06682130	Glofitamab	R/R DLBCL	Possible glofitamab bridging before ASCT if MRD-positive after 2L salvage

BsAb: bispecific antibody. MRD: measurable residual disease. 1L: frontline. DLBCL: diffuse large B-cell lymphoma. R-CHOP: rituximab, cyclophosphamide, doxorubicin, vincristine, and prednisone. CAR-T: chimeric antigen receptor T-cell therapy. R/R: relapsed/refractory. Pola-R-mini-CHP: polatuzumab vedotin, rituximab, dose-attenuated cyclophosphamide, doxorubicin, and prednisone. ctDNA: circulating tumor deoxyribonucleic acid. 2L: second-line. ASCT: autologous stem cell transplant.

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
