# Peer review of "Measurable Residual Disease Testing During Treatment with Bispecific Antibodies for Lymphoma"

_cancers, 2025, doi:10.3390/cancers17071153_

Round 1
Reviewer 1 Report
Comments and Suggestions for Authors
in this ms Authors are reviewing data on the value of MRD assesment at ctDNA level in Lymphoma patients treated with bispecifics Moads. The article is well written and very informative on this subject
Major Comments
- There are up-dated results from many studies in the latest ASH meeting so some data must be updated and enriched. Especially data on sustained MRD ( ex after stop of Glofitamab) should be mentioned
- Comments about possible future value of MRD assesment and MRD tailored therapies should be included. Especially when comparing Fixed duration treatment(glofitamab) and continoous treatment (epcoritamab-can we stop and reintroduce EPCO?) in R/R DLBCL
- Statements that for now there are not enough data for MRD guided therapies must be very clear and a statement that all data are coming from non-randomized studies is merritable. Furthermore MRD assesment is made on selected populations of each one of the studies and with differents protocols and time points of assesment making data difficult to interprete. Authors must make some personal suggestions on this issue
- A list of currently running trials with MRD tailored therapies in Lymphoma with bispecific Ads will be of value to be included
Reviewer 2 Report
Comments and Suggestions for Authors
The manuscript gives an overview on the methodological approaches and the clinical significance of measurable residual disease (MRD) in patients with B-cell lymphoma undergoing Bispecifica Antibody (BsAbs) therapy.
This is an area of emerging interest, and the review offers well documented information on MRD assessment and its prognostic value. Thus, the review may represent a useful tool for the most accurate management of high-risk B-cell lymphoma patients.
To widen the presentation, I believe that some additional issues should be mentioned and somehow discussed:
- The various biological sources for MRD assessment should be clearly described including the novel circulating analytes such as tumor-educated platelets and extracellular vesicular DNA. In addition, sampling method, preservation technique, and sample transportation can all affect MRD detection. Indeed, the quantity and quality of the evaluated samples is crucial for guaranteeing the accuracy and reliability of MRD assessment. This should be clearly stated and discussed
- For certain B-cell lymphoma, namely Follicular (FL), Mantle-cell (MCL) and small-lymphocytic/chronic lymphocytic leukemia (SLL/CLL), residual tumor cells may circulate in the peripheral blood. Flow cytometry has been employed since long time for the detection of these tumor B-cells. The possible use of flow cytometry for MRD detection should be mentioned with advantages and disadvantages compared to modern NGS-based approaches
- Qualitative and quantitative PCR (RQ-PCR) have paved the way of MRD assessment in B-cell lymphoma. This should be clearly acknowledged. In addition, the use of RQ-PCR along with the more recently developed digital droplet PCR (ddPCR), offer sensitive quantitative detection of MRD when tumor-specific somatic mutations are available. Again, this approach should be mentioned and discussed in comparison with NGS
- Whole Genome Sequencing (WGS) is an alternative approach for DNA-tumor detection. WGS does not target specific mutation sites, but instead it recognizes copy number aberration (CNA) or copy number variation (CNV). Their profiles have been utilized in several studies involving both Hodgkin’s and non-Hodgkin B-cell lymphoma. This should be quoted and discussed in comparison with NGS-based methods
- The Authors remark on the use of MRD to guide changes in treatment. This is a relevant issue. However, the question of maintenance therapy is not clearly cited. Indeed, in FL and in MCL as well, the maintenance therapy with anti-CD20 MoAb is effective, although its use is viewed with concern in some situations due to the infection risks associated with prolonged immunosuppression. Information about MRD following induction/consolidation may help in the decision about the possible treatment extension with maintenance. Some recent reports on this issue should be quoted and discussed
